# Angle Assessment for Upper Limb Rehabilitation: A Novel Light Detection and Ranging (LiDAR)-Based Approach

**DOI:** 10.3390/s24020530

**Published:** 2024-01-15

**Authors:** Luan C. Klein, Arezki Abderrahim Chellal, Vinicius Grilo, João Braun, José Gonçalves, Maria F. Pacheco, Florbela P. Fernandes, Fernando C. Monteiro, José Lima

**Affiliations:** 1Research Centre in Digitalization and Intelligent Robotics (CeDRI), Polytechnic Institute of Bragança, 5300-252 Bragança, Portugal; luanklein@alunos.utfpr.edu.br (L.C.K.); arezki@ipb.pt (A.A.C.); viniciusgrilo@ipb.pt (V.G.); jbneto@ipb.pt (J.B.); goncalves@ipb.pt (J.G.); pacheco@ipb.pt (M.F.P.); fflor@ipb.pt (F.P.F.); monteiro@ipb.pt (F.C.M.); 2Department of Electronics (DAELN), Universidade Tecnológica Federal do Paraná (UTFPR), Campus Curitiba, 80230-901 Curitiba, Brazil; 3School of Science and Technology, Universidade de Trás-os-Montes e Alto Douro (UTAD), 5000-801 Vila Real, Portugal; 4Faculty of Engineering, University of Porto (FEUP), 4200-465 Porto, Portugal; 5Institute for Systems and Computer Engineering, Technology and Science, INESC TEC, 4200-465 Porto, Portugal; 6Laboratório para a Sustentabilidade e Tecnologia em Regiões de Montanha (SusTEC), Instituto Politécnico de Bragança, 5300-252 Bragança, Portugal

**Keywords:** join angle measurement, Artificial Intelligence, motion capture, LiDAR, robotic rehabilitation

## Abstract

The accurate measurement of joint angles during patient rehabilitation is crucial for informed decision making by physiotherapists. Presently, visual inspection stands as one of the prevalent methods for angle assessment. Although it could appear the most straightforward way to assess the angles, it presents a problem related to the high susceptibility to error in the angle estimation. In light of this, this study investigates the possibility of using a new approach to angle calculation: a hybrid approach leveraging both a camera and LiDAR technology, merging image data with point cloud information. This method employs AI-driven techniques to identify the individual and their joints, utilizing the cloud-point data for angle computation. The tests, considering different exercises with different perspectives and distances, showed a slight improvement compared to using YOLO v7 for angle calculation. However, the improvement comes with higher system costs when compared with other image-based approaches due to the necessity of equipment such as LiDAR and a loss of fluidity during the exercise performance. Therefore, the cost–benefit of the proposed approach could be questionable. Nonetheless, the results hint at a promising field for further exploration and the potential viability of using the proposed methodology.

## 1. Introduction

The measurement and identification of the patient’s Range of Motion (ROM) is an essential element for assessing and diagnosing shoulder pathology on one hand but also, on the other hand, a pivotal point to monitor the reliability of the rehabilitation procedure [1]. Different methods are applied to assess rehabilitation procedures, including clinical assessments [2], imaging techniques [3], and wearable devices [4].

The current standard in medical practice for assessing a patient’s angles relies on visual inspection, which is often regarded as an approximate method and can be complicated due to the dynamic nature of the patient’s mobility throughout the rehabilitation process [5,6]. Some studies have shown that visual examination can present errors up to 10º [7] and concluded that they have poor reliability [8]. An alternative solution includes using a goniometer [5]. However, this tool has a high level of variance and is often difficult to use because professionals are not familiar with its use [8,9].

On this basis, there is a necessity for new approaches that provide more accurate estimates. Some studies have been developed to bring considerable improvement, for example, using images. However, there is still room for improvement, and the possibility of using more precise sensors than images is an interesting option.

The present work is a preliminary study on the feasibility of using Light Detection and Ranging (LiDAR) for joint angle estimation in rehabilitation. This new approach uses an Artificial Intelligence (AI) framework that enables pose detection in an image, and by fusing image and point cloud data, the system can better estimate the angle between the joints. Therefore, this study aims to contribute with a more accurate method to calculate the angles during rehabilitation sessions, as facilitated by a robotic arm and controlled by SmartHealth software 0.0.9 [10,11]. The presented results aim to assess the effectiveness and reliability of the proposed method in robotic rehabilitation and its limitations for different views and distances.

In addition to this Introduction, this paper is divided as follows: Section 2 presents related works. Section 3 outlines the methodology for exploiting LiDAR point clouds for angle detection in SmartHealth software. Section 4 highlights the obtained results and offers some details for additional discussion. Finally, Section 5 provides a general conclusion about the work and the proposition of future developments that will enhance this study.

## 2. Related Work

Throughout this section, a non-exhaustive exploration of the techniques employed for the detection of a person’s angles in the medical environment is carried out. It has been observed that there is a general trend towards the use of AI for remote wireless solutions and Inertial Measurement Units (IMUs) as a wearable solution.

### 2.1. Artificial Intelligence and Depth Cameras

A large majority of the works cited mention the application of AI for this purpose, for instance. A squat angle assessment technique was investigated in the study presented in [12], and a mean absolute error (MAE) of 8.64° was reported. Based on a single RGB camera placed in front of the subject, the system continuously provided the angle of the subject’s knee at a rate of 25 frames per second (FPS). The training was carried out using two cameras, one at the front and the other at the side. The actual angle used to train the algorithm was deduced offline using an algorithm that was not mentioned. CHHOEUM et al. [13] introduced a Convolution Neural Network (CNN) designed to assess knee joint angle based solely on mapping foot pressure. Three types of footwear were involved in the experimentation. The algorithm’s performance was evaluated by comparing the results obtained from reflective markers auto-tracked using Kinovea software. The reported accuracy of this system comprises was 70% to 90%.

In another study [14], a system consisting of two RGB-D cameras was employed to ascertain the three-dimensional (3D) positions of the subject’s joints. While the study did not explicitly mention the implementation of AI, it is reasonable to infer its application, given that the estimation of joint position typically involves computer vision and AI methodologies. The performance of this system was juxtaposed with that of a photonic-based optical fiber (POF) system and an IMU system. The investigation revealed that non-portable systems still exhibit notable errors in motion analysis and subsequently proposed post-processing techniques to mitigate these inaccuracies. Hii, Chang Soon Tony et al. [15] conducted a comparative analysis of different pose-estimation techniques for lower limb and gait rehabilitation, including OpenPose, Mediapipe, and MMPose, employing two cameras positioned alongside the test pathway. The study relied on reflective markers to establish reference angles, and subsequent computations were executed through the Quintic Biomechanics software. It was concluded that Mediapipe was suitable for this case study.

The authors also investigated, in a previous study [16], the application of AI in this application area. The results of two vision-computer approaches, Mediapipe and YOLO v7, were compared. The study concluded that YOLO v7 could be applied to this type of application, where errors of less than 8° (and 5° on average) were observed for the exercises studied. In contrast, Mediapipe performed poorly, with errors of up to 25° for most of the exercises studied.

### 2.2. IMUs

A strong tendency to use IMUs for angle estimation has also been observed. In [17], a smartphone application using the smartphone’s built-in sensors was proposed to estimate the angle of subjects. This tool was tested both in a controlled environment and a real environment. The reported results showed low assessment errors. Another study in [5], focusing on the lower limb, proposed a Machine Learning (ML) technique for angle assessment of IMUs’ output data. An investigation of applying one or a combination of two IMUs has been proposed to capture the subject’s angle evolution. A CODA motion capture system is also applied as a ground truth. The results indicate no differences between the usage of one and two IMUs, with a reported Root Mean Square Error (RMSE) of 4.81°.

In [18], a wireless sensor network based on two Shimmer devices is reported to monitor the patient’s movement in real time; these IMUs; network accuracy is briefly compared with a goniometer, with a difference between the two measurements reaching no more than 3°. Lee, Wang Wei, et al. [19] also proposed a wireless system composed of seven sensor nodes in direct communication with a central smartphone application. Clinical trials of this system have been reported in [20]; 19 healthy subjects and 20 disabled patients have participated in the study, where the system has been compared with a goniometer. A high correlation between the goniometer and the system has been reported.

Alternatively, research reported in [21,22] has proposed a wearable solution based on liquid metal sensors to capture the angle and general motion of the ankle complex. The study’s findings affirm a high confidence level in the results obtained from the device, with an RMSE of up to 3.16°, compared to a 3D motion capture system used as ground truth. Another approach, described in [23], proposes the application of a flexible polymer fiber (POF) optic sensor to assess the angle of the human joint. Tests were carried out on the elbow and knee joint with an RMSE of 1.5°.

Other research teams propose a fusion between systems, for example, in [24], where a hybrid system combining POF and IMU for lower limb assessment was presented. The angular velocity measured using the BN055 gyroscope is used to compensate for the hysteresis effect of the POF sensors, and the data obtained from both sensors are merged and filtered using a Kalman filter. The angle estimated using the device was compared with the encoder of an exoskeleton used for gait rehabilitation, resulting in an RMSE of less than 4°.

To the best of the authors’ knowledge, only one ongoing study proposes the application of a LiDAR camera for angle assessment. As of the time of writing, this study remains unpublished and is currently undergoing review, with further details available in [25]. This study harnesses a fixed LiDAR camera with Cubemos and Mediapipe for human joint estimation. The approach was validated by comparing the results obtained from a Mocap system. Thirteen participants assessed a single exercise. The preliminary study findings highlight the effectiveness of Cubemos over Mediapipe, as it yields an MAE of less than 10°. Nevertheless, the proposed system exhibits limitations when capturing movements directed toward the camera. This limitation underscores the importance of optimal camera positioning for accurate angle assessment.

## 3. Methodology

### 3.1. Data Collection

In order to collect data, an Intel RealSense D415 RGB camera was employed, synchronized with an RS16 LiDAR (Manual available at: https://cdn.robosense.cn/20200723161715_42428.pdf, accessed on 10 December 2023). Both of these devices were attached to the Universal Robot 3 (UR3) robotic arm (Manual available at: https://s3-eu-west-1.amazonaws.com/ur-support-site/32341/UR3_User_Manual_en_E67ON_Global-3.5.5.pdf, accessed on 12 December 2023), as depicted in Figure 1, which shows the system implementation.

Specific software was developed to ensure synchronized interaction between the sensors mentioned above and the robot. It was also responsible for gathering and saving the collected data. The code was programmed in Python 3.8.10, with OpenCV 4.8.0, Ubuntu 20.04, and the Robot Operating System (ROS) Noetic.

In summary, the RS16 LiDAR and the D415 Camera had their respective topics on the ROS, where they continuously published the gathered data. Another function was responsible for reading the topics and saving the data. In addition, the same software was responsible for connecting to the UR3 and obtaining its current position. With all the information, it was possible to save the data systematically. To ensure the synchronization between all the data, it was only saved when all the data were available; i.e., the image, the point cloud, the robot pose, and the IMU data were all available simultaneously.

As the RS16 LiDAR is a 360° sensor, its capacities were limited to capturing only the participant’s movement. An angular span, $\pm \phi_{max}$, was chosen as ±18°, with a maximal detection distance of 4.8 m.

The camera was strategically positioned atop the LiDAR to ensure that both instruments captured coinciding data fields. Even though minor discrepancies between their data might occur, this arrangement facilitates easier data integration. The LiDAR was oriented vertically to optimally utilize its 16 light beams, enhancing the detection of the patient. For comprehensive participant identification, both the LiDAR and the camera were mounted on the UR3. This setup allowed the UR3 to execute solely horizontal motion, shifting between −0.3 m to −0.1 m on the x-axis during the data acquisition. Figure 2 depicts the system’s implementation and highlights the overall relation between each used component.

The designated points from (1) to (4), outlined in the previous figure, correspond to the link between the system’s component and the computer. They can be defined as follows.

Corresponds to the data collected and transferred from the ground truth to the computer. This communication operates via Bluetooth. The data related to the angle of the joint used in the movement are sent as a character string and subsequently archived within a designated *.txt* file for later analysis.Corresponds to the images captured by the RealSense camera. These images are saved in the *.jpg* format, allowing for convenient access and reference throughout the study.Corresponds to the point cloud captured by the RS16 LiDAR and sent to the computer. The raw point cloud data are saved in a structured *.bag* format.Associated with the communication between the robotic arm and the computer, the data sent represent the position of the robot by encompassing six parameters (*x*, *y*, *z*, $\theta_{x}$, $\theta_{y}$, $\theta_{z}$), which represent the end effector’s position. The data are sent as a character string via a Transmission Control Protocol/Internet Protocol (TCP/IP). The data are stored upon receipt in a dedicated *.txt* file.

### 3.2. Data Processing and Angle Calculation

Several steps were required to obtain the value of the angle. Figure 3 presents the flowchart of the needed steps, starting with the data collection, which was explained before, and finishing with the angle calculation. Each step is detailed in sequence, giving further details.

#### 3.2.1. Fuse the Points in the Point Clouds

Once the data were available, i.e., once the data were collected, the first processing step was the fusion between different point clouds. Initially, it is necessary to rotate the point clouds and the images collected since the LiDAR and the camera are rotated by a 90° angle during the collection. To perform the rotation of the point cloud, the $Y_{L}$ and $Z_{L}$ axes, related to the LiDAR frame as presented in Figure 1, were mutually swapped.

In sequence, since several collections were performed for the same person’s pose, it is necessary to merge all the point clouds into only one. The fusion methodology was to select a reference to the measurements and make the fusion of all the other measurements based on the reference. In practical terms, the process was as follows. Select the first measurement as a reference, and, through an interactive loop, make the fusion measurement-by-measurement, adding a horizontal offset defined by the horizontal distance between the first measurement and the other measurements (collected from the robot’s position, which was holding and moving the LiDAR and the camera). For instance, the measurement at iteration *n* is selected as the reference. Measurement n+1 is merged with the reference with a horizontal offset, measurement n+2 is merged with the previous measurement (n+1) at another horizontal offset, and so on until measurement n+10. This process is illustrated in Figure 4, where Figure 4A represents the cloud point related to the first measurement; Figure 4B the *n* measurement merged with n+1; Figure 4C the merge between measurement n,n+1,n+2; Figure 4D the merge of n,n+1,n+2,n+3; and, finally, Figure 4E the final merge between all the 11 measurements (n,n+1,⋯,n+10), where a different color represents each one. The choice to use 11 measurements was due to the whole range of the data collection; i.e., a complete horizontal movement of the UR3 was defined to collect 10 measurements. However, due to delays, in some cases, 11 were collected. So, the choice was to use 11 because if the collection achieved 11 measurements, all those could be used, and if only 10 were collected, using 11 meant that one of the measurements was replicated. However, this was not a problem because the points could only be overlapped without impacting the approach.

#### 3.2.2. Identify the Person in the Image

The second processing step is the person’s identification in the image. The tool used for identification was YOLO v7, which is one of the state-of-the-art tools in object detection in images [26]. The pre-trained model with the weights used in this work is called *yolov7-w6-pose.pt* (https://github.com/WongKinYiu/yolov7, accessed on 7 October 2023), which is the same used in the previous work of the authors [16]. With this framework, a bonding box was defined around the person, with the position of the edges defined in pixels. With these edges, it was possible to define the size of the person (height and width) in pixels. It is important to emphasize that since the data collection with the LiDAR only collected data from the upper part of the body, as presented in Figure 4, the person detection was also adjusted to identify only the upper part of the body. So it was empirically defined which part of the image should be considered and which should not, and then, to the person’s identification with YOLO v7, a black rectangle was put over the image to hide the part of the body that LiDAR did not collect. Later, after the person’s identification, the black rectangle was removed. An example of person identification using the YOLO v7 framework is presented in Figure 5, where a green rectangle was drawn around the upper part of the person.

#### 3.2.3. Fit the PointCloud in the Person Identified

In sequence, a copy of the point cloud was performed, and the depth of the points was unconsidered; i.e., at this moment, all the points were in 2D. The points were resized according to the person’s size: the height of the point cloud must be the same as the person in the image and the same for the width. This was carried out by encountering the proportion between the sizes (point cloud and person), and the position of the points in the point cloud was multiplied by the defined proportions. Once both had the same size, the point cloud was fitted on the image to be on the person at the same position. An example of this is presented in Figure 6, where the white points are the point cloud in 2D, with the adaptation to fit in the person on the image. It is important to emphasize that they may be flawed, as the person identification may present limitations. Indeed, it has been observed that for some images, certain parts of the body, such as the hand, may be ignored.

#### 3.2.4. Identify the Person’s Joints

The following step is the person pose identification, aiming to identify the body’s joints. Through the YOLO v7 application, the framework returns the position of each joint in pixels. The procedure applied to this was similar to the one present in [16].

#### 3.2.5. Identify the Nearest Point of Each Joint

Once the joint position is known in a pixel, the system searches for the nearest points for each joint in the point cloud. The Euclidean distance was considered, and in a 40-pixel radius, the closest 10 points were selected. If fewer than 10 points were within this radius, all available points were selected. The choice of these values was made empirically. Since the objective of this study is only to validate the possibility of using the proposed approach, these values can be further explored in future works, aiming to find an optimal value and improve the quality of the estimations.

#### 3.2.6. Obtain the Original Value of Each Point Identified

Once the points were defined, they were described according to the index, and through the index, the original points in the point cloud were obtained. An example of the joint identification and point definition is presented in Figure 7, where the white points are the point cloud, the green points are related to the right wrist, the red points are related to the right shoulder, and the blue points are related to the right hip.

#### 3.2.7. Calculate the Angle with the Original Values

Upon the point selection related to each joint and the respective recovery of their original values, each joint’s mean coordinates were calculated, yielding a singular representative point denoted by coordinates (x,y,z), referred to as the centroid. This centroid, in turn, serves as the basis for further computations involving angular measurements through the application of a defined mathematical formula as follows:(1)$$\theta =\arccos\left(\frac{\mathrm{BA}\cdot \mathrm{BC}}{∥\mathrm{BA}∥\cdot ∥\mathrm{BC}∥}\right),$$
where *A*, *B*, and *C* are the centroids of each joint; $\mathrm{BA}=(x_{a}-x_{b},y_{a}-y_{b},z_{a}-z_{b})$ is the vector with endpoints points *A* and *B*; and $\mathrm{BC}=(x_{c}-x_{b},y_{c}-y_{b},z_{c}-z_{b})$ is the vector whose endpoints are *C* and *B*.

Figure 8 presents an example of execution of the elbow flexion exercise ($R_{7}$), which will be explained further in Section 3.3, considering a distance of 4 m and a oblique perspective. Each column represents one pose performed during the exercise, with the first row showing the participant’s picture, the point cloud fitted on the person, and the colored joint points. The second row shows the original point cloud in 3D.

### 3.3. Rehabilitation Exercises

The arm is a complex part of the human body, composed of several joints, including the wrist, elbow, and shoulder. Each joint is composed of several rotation axes, enabling the different movements of the arm. In the rehabilitation process, the exercises are performed in countless repetitions, aiming to exercise and develop the movement of the joint to be as natural as possible. Further details about these exercises are out of the scope of this study, and interested readers are referred to the following references [16,27].

The subject was positioned in front of the LiDAR, at distances of 3, then 4 m, as previously represented in Figure 2. Three exercises were performed for this study by only one healthy human test subject, utilizing his right arm. During the trials, the participant performed several poses, both in a frontal view (Figure 9), a lateral view (Figure 10), and an oblique view (such as in Figure 8). Figure 9 demonstrates the targeted human joint for this study.

The exercise depicted in Figure 9A represents the shoulder abduction movement relative to the $R_{A}$ rotation axis. The subsequent test shown in Figure 9B might seem similar to the previous one. However, it is based on the shoulder flexion exercise, a movement related to the $R_{F}$ rotation angle. The last targeted movement (Figure 9C) represents elbow flexion ($R_{7}$), with the center axis located in the elbow. All of the exercise representations above demonstrate the Point of View (POV) of the camera and are referred to in this study as the frontal view. According to the proposed system, the same test series is also performed in a lateral view, as demonstrated in Figure 10, and in an oblique view.

For each of the targeted exercises, the participant proceeded as follows:The participant initiated the exercise by moving their hand;He aimed to maintain a relatively fixed position;The participant waited for several cycles to complete the robotic arm’s movement;After the cycle concluded, the participant moved their arm to another position for further assessment.

This approach offers the possibility of investigating LiDAR’s capabilities and limitations in a continuous and discontinuous subject motion. The limitation encountered will be further discussed in the Results and Discussion section.

### 3.4. Ground Truth System

In order to assess the quality of the results obtained throughout the proposed approach, it was necessary to obtain the real value of the angle during the execution of the exercise. These real values are called ground truth values and will be used as the reference to the results obtained through the proposed approach. To obtain the real values with a high level of confidence, an external measurement method was developed, and this section presents further details about it. Once the system was developed, it was executed in parallel to the data collection with the LiDAR in all exercises performed: shoulder abduction ($R_{A}$), shoulder flexion ($R_{F}$), and elbow flexion ($R_{7}$). Once the angle values from the ground truth and the proposed system were obtained, the metrics for the evaluation were calculated (the details about the metrics and the evaluation method are presented in Section 4). The ground truth system developed consists in a device that was developed based on Bosch’s BNO055 IMU (manual available at: https://cdn-shop.adafruit.com/datasheets/BST_BNO055_DS000_12.pdf, accessed on 25 November 2023). The device was developed using an ESP32, the IMU sensor previously mentioned, and a pair of 18,650 batteries to power the system. The communication between the device and the computer that collected the data was via Bluetooth, as depicted in Figure 2.

The IMU used contains three motion sensors (gyroscope, accelerometer, and magnetometer) and a microcontroller running an algorithm that fuses the sensors, filters out unwanted measurement noise, and communicates with external devices, such as other microcontrollers. Unlike other commercial sensors, the algorithm for fusing and processing the sensor signals must be implemented on the system’s main microcontroller. This brings an advantage to the system since the algorithm embedded in the sensor was developed by the manufacturer and is adapted and developed, especially for this model. Figure 11, shows the wearable IMU attached to the UR3.

To assess the ground truth accuracy, the device was compared along the UR3 robotic arm. The UR3 is a versatile arm that performs the movement with minimal errors, making the ground truth validation process more cohesive [4]. The accelerometer was attached to the arm shown in Figure 11. Only the robotic arm elbow joint was set to move. The robot performed a sinusoidal movement first, from 0° to 90°, at the maximum robot speed. Then, the robot moved from 0° to 90°, stopping at some angle (20, 30, 50, 70) for 5 s. Figure 12 compares the robot’s current elbow angle along the proposed ground truth measurement.

It can be observed from the above figure that the fitting between the two measures. An MAE of 1.55° has been computed, which makes the proposed IMU a viable system to be applied as a ground truth. However, limitations must be pointed out, as it is critical to position the sensor perfectly in alignment with the arm and ensure that it remains fixed throughout the tests.

## 4. Results and Discussion

Following the data-collection procedure for the exercises presented above, the data were processed, and the results are presented in the following section. Each sub-section refers to a type of exercise. The system’s effectiveness was tested from different perspectives (frontal, lateral, and oblique views) and different distances (3 and 4 m).

Three approaches were compared, YOLO v7, as well as two other approaches proposed in this study:LiDAR + AI Simple (LiDAR + AI S), using YOLO v7 for participant identification, coupled with data collected using LiDAR by a single measurement.LiDAR + AI Full (LiDAR + AI F), similar to LiDAR + AI S, but using 11 measurements to obtain an ideal mapping of the subject’s arm.

In the following section, several graphics are provided with the aim to evaluate the efficiency of the studied approaches, organized according to the performed exercises (shoulder abduction ($R_{A}$), shoulder flexion ($R_{F}$), and elbow flexion ($R_{7}$)). Each graphic presents six plots organized in two lines and three columns. The first line presents the results at a 3-m distance, while the second line presents the results at a 4-m distance. The columns from left to right indicate the results acquired at frontal, oblique, and lateral views. The abscissa indicates the number of iterations the approaches underwent, where one iteration indicates one sample taken from the approaches. In addition, the ordenada values are represented in degrees. It is important to note that the data-acquisition sampling frequency was set to approximately 10 Hz.

In the plots, the blue line series denotes the ground truth of the arm angle, and the former system was elaborated in Section 3.4. The YOLO v7 and the study’s proposed frameworks (LiDAR + AI S and LiDAR + AI F) are represented by the yellow, purple line series and the black **x**’s, respectively. As previously mentioned, each subplot’s distinctive black **x**’s denote the second proposed framework executed with eleven sampling iterations. This representation was chosen due to the discontinuities of estimating at every eleventh iteration.

In addition, for a comprehensive understanding of the LiDAR + AI S method, the graphical representation reveals a distinct grey dotted line, denoting the calculated average of predicted points generated, calculated for each angle step separately.

Moreover, some metrics are necessary to evaluate the quality of the approaches and enable a comparison between them in terms of estimating angles. A combination of MAE and RMSE is suggested to evaluate the estimations [5]. In addition to these, other metrics can also be used to contribute to the system’s evaluation, such as Mean Absolute Percentage Error (MAPE), Normalized Root Mean Squared Error (NRMSE), Sample Standard Deviation (STD), and $R^{2}$ [28]. These metrics were computed by applying the sklearn.metric (version 1.2.1) library. On the other hand, the Numpy (version 1.23.5) was also applied to calculate the sample standard deviation. The results can be consulted in Appendix A.

### 4.1. Shoulder Abduction ($R_{A}$)

Figure 13 presents the summary of results from the shoulder abduction exercise, with the angle estimated using the YOLO v7 and the LiDAR approaches, compared to the ground truth.

In the frontal view scenario, it is evident that optimal results were acquired at a 4-m distance. YOLO v7 demonstrated impressive performance, achieving an MAE of 3.58°, an RMSE of 4.07°, and an STD of 2.00° at 3 m. This performance is further elevated at a 4-m distance, with an MAE of 0.98°, an RMSE of 1.26°, and an STD of 1.17°. The LIDAR + AI S and LIDAR + AI F models also exhibited precise angle estimation. For instance, at a 4-m distance, they report MAEs of 4.42° and 2.68°, RMSEs of 5.00° and 2.77°, and STD values of 3.47° and 2.49°, respectively.

Both the proposed approaches excel for the oblique view, with a slight advantage observed at 3 m. In this scenario, the LiDAR + AI S consistently provided angle estimation within a range of ±6°, while the LiDAR + AI F the range was ±4°. At 4 m, the LiDAR + AI F provided a satisfactory accuracy ranges of 2.41° ± 2.39°.

All approaches provide relatively low MAE, RMSE, and MAPE values for both the frontal and oblique views, indicating good accuracy. However, a different scenario unfolds in the lateral view, where all approaches encounter challenges in accurately tracking angles during lateral movements. In this regard, none of the approaches can be proposed as a valuable method. Figure 14 provides a summarized representation of the MAE, mean, and STD for all of the investigated tests related to $R_{A}$ rotation.

### 4.2. Shoulder Flexion ($R_{F}$)

Figure 15 offers an overview of the obtained results for the shoulder flexion ($R_{F}$) rotation.

When analyzing Figure 15, which represents the results according to the shoulder flexion ($R_{F}$), one can note that the LiDAR + AI F, in general, outperforms the other approaches in terms of MAE, RMSE, and MAPE at a distance of 3 m. At the same time, LiDAR + AI S has metrics where it outperforms the other approaches at a 4-m distance. As for the oblique and lateral views, the same observation is made, where the LiDAR + AI F outperforms the other methods consistently. The other proposed approach, LiDAR + AI S, has some metrics that perform better than the third-party approaches and others that underperform them. Figure 16 provides a summarized representation of the MAE, mean, and STD for all of the investigated tests related to $R_{F}$ rotation.

### 4.3. Elbow Flexion ($R_{7}$)

With its three perspectives, the last exercise is represented in Figure 17, wherein the frontal view, LiDAR + AI F, and YOLO v7 approaches perform better by a significant range against the other approach despite the distance. Regarding the isometric view, the proposed approaches perform consistently better than the other approach, whereas, in some metrics, the LiDAR + AI S performs better. In others, the LiDAR + AI F performs better. Finally, the worst performance for all tests and exercises is once again the lateral view for the elbow flexion. This is expected where the least number of informative data can be acquired, i.e., most of the acquired data are ambiguous, increasing the error metrics.

Figure 18 provides an overview of the Mean, MAE, and STD for each approach at different distances and views.

In addition, the overall algorithm accuracy for all the approaches, distances, exercises, and views (excluding lateral view) was investigated. The error distribution was obtained and is shown in Figure 19.

Based on the comprehensive test results, it is evident that the distance does not significantly influence the quality of results obtained with the YOLO v7 approach. In contrast, for the LiDAR + AI S approach, observations reveal a nuanced impact of distance, with marginally improved results at a 3-m distance compared to a 4-m distance in most scenarios.

Moreover, substantial variations in accuracy become apparent across different viewing angles and test conditions, particularly for the LiDAR + AI S approach, where a higher degree of variance is observed during elbow flexion. This variance can be attributed to the relatively lower probability of detecting points in a single iteration. However, the LiDAR + AI F approach remains unaffected by this variation due to its ability to perform a complete arm scan, consequently enhancing prediction quality.

A noteworthy observation is the resemblance between the approaches. In the best-case scenarios, the LiDAR approach exhibits a minimal advantage over YOLO v7, with an angular difference of less than 1° in frontal views. Conversely, LiDAR-based approaches consistently outperform YOLO v7 when considering oblique views, with the LiDAR + AI F approach particularly noteworthy for consistently predicting angles below 10° in most cases.

## 5. Conclusions

This study aimed to investigate the feasibility and the potential application of a LiDAR-based approach dedicated to upper-limb angle detection. Both of the proposed approaches involve the collection and utilization of point cloud data from a LiDAR device. Subsequently, a Machine Learning technique and its various joint points are applied for subject identification. The point cloud data are then integrated into the person. The corrected joint coordinated is then utilized to calculate the targeted joint.

A ground truth system was concurrently developed to facilitate the assessment of the proposed approach’s accuracy. In addition, both approaches were also compared to YOLO v7, which has already demonstrated good abilities in this matter. In total, 18 different poses were evaluated, encompassing three exercises ($R_{A}$, $R_{F}$, and $R_{F}$) from three different perspectives (frontal, oblique, and lateral views) and at two different distances (3 and 4 m).

One notable observation from the results is the slight enhancement of prediction accuracy, achieved by utilizing multiple measurements (LiDAR + AI F) for the different performed rotations, especially at an oblique view. However, this slight increase in accuracy comes at the cost of a decrease in prediction fluidity, where the approach requires 11 iterations for prediction. This leads to a more discontinuous process that requires the person to remain in a fixed position during data acquisition. The LiDAR + AI S also showed promising results for $R_{A}$ and $R_{F}$, where there are higher chances to encounter the hand at any iteration, in contrast with the $R_{7}$ where, in some iterations, not enough body points are captured with the LiDAR, thus increasing variance.

Undoubtedly, both of the proposed approaches introduce innovative methods for wireless angle calculation. However, it is essential to acknowledge certain drawbacks associated with the proposed approach. The primary concern pertains to the system’s costs, as the LiDAR sensor considerably increases the overall system cost, making it much more expensive than the YOLO v7 sole requirement of a standard camera.

A further in-depth investigation could explore alternative frameworks for person and joint identification, as well as the incorporation of vertical, angular, and in-depth robot movement for a complete person scan. Furthermore, the normal distribution depicted in Figure 19 not only underlines the robustness of our experimental results but also paves the way for potential improvements, where it might be considered a fusion between the LiDAR + AI S approach and the YOLO v7’s output angle prediction to further improve estimation quality.

## Figures and Tables

**Figure 1 sensors-24-00530-f001:**
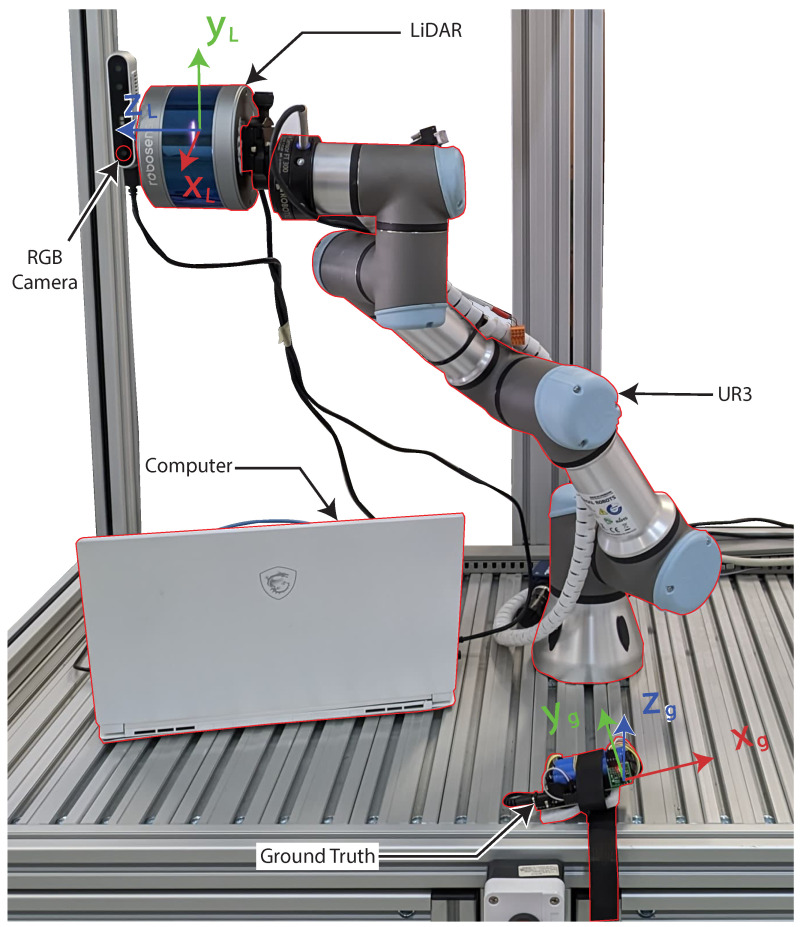
Proposed system, representing the different coordinate frame.

**Figure 2 sensors-24-00530-f002:**
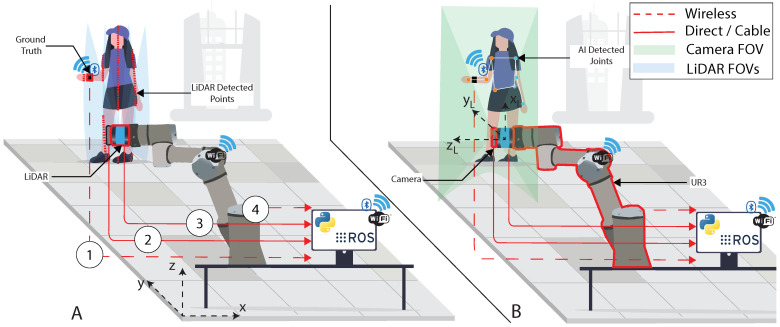
Example of the position of the sensors and the patient during the data collection. (**A**) LiDAR field of view. (**B**) Camera field of view.

**Figure 3 sensors-24-00530-f003:**
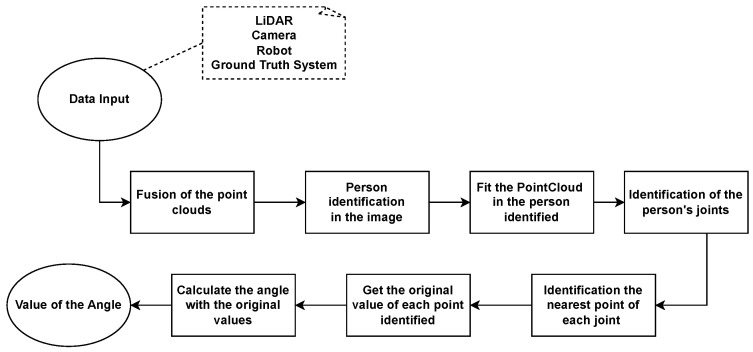
Flowchart of the proposed approach.

**Figure 4 sensors-24-00530-f004:**
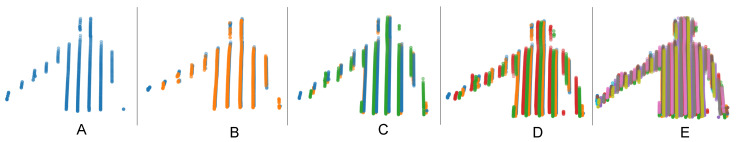
Example of the LiDAR measurement process merge: (**A**) the initial measurement, (**B**) the fusion between the initial measurement and the second, (**C**) fusion with the third measurement, (**D**) fusion with the fourth, and (**E**) the final fusion of the points cloud, with each color representing one measurement.

**Figure 5 sensors-24-00530-f005:**
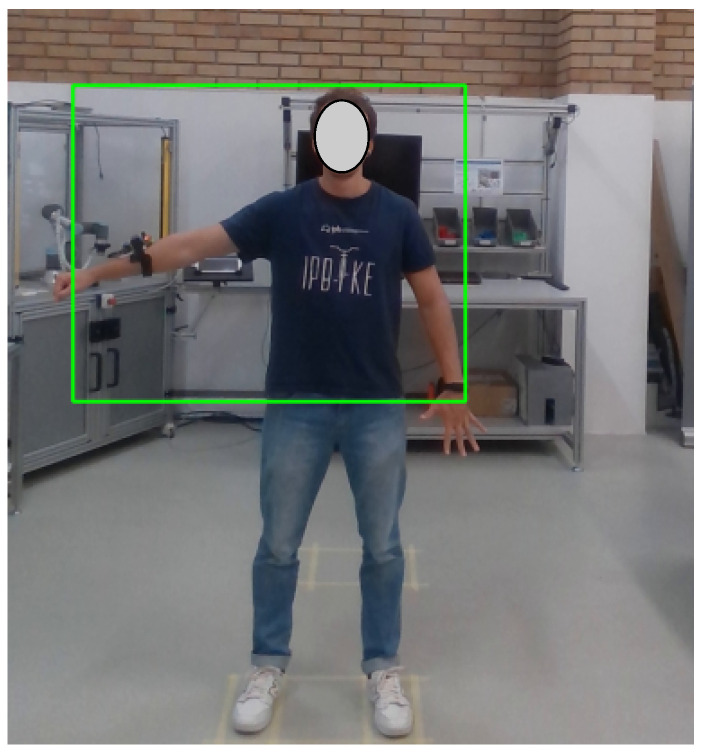
Example of person identification using YOLO v7.

**Figure 6 sensors-24-00530-f006:**
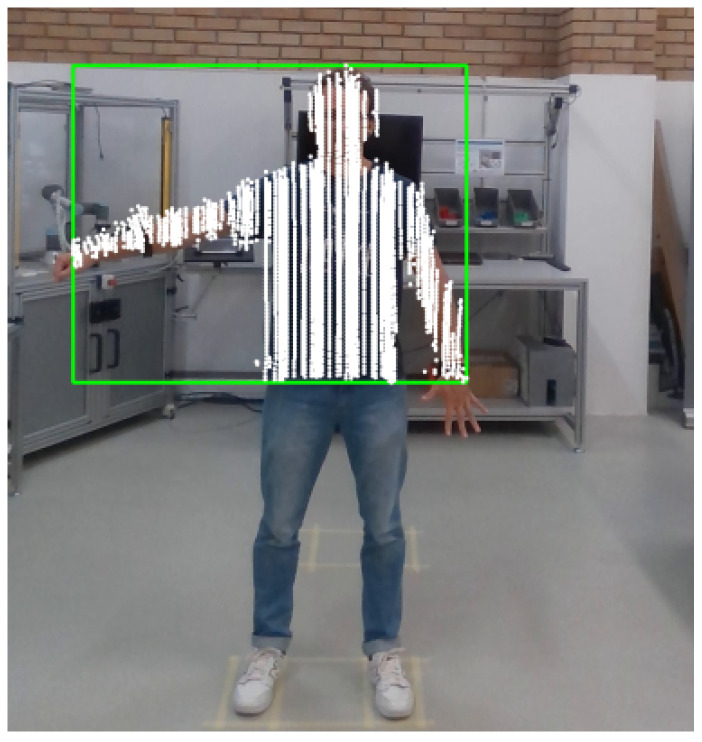
Example of the point cloud fitting in the person on the image.

**Figure 7 sensors-24-00530-f007:**
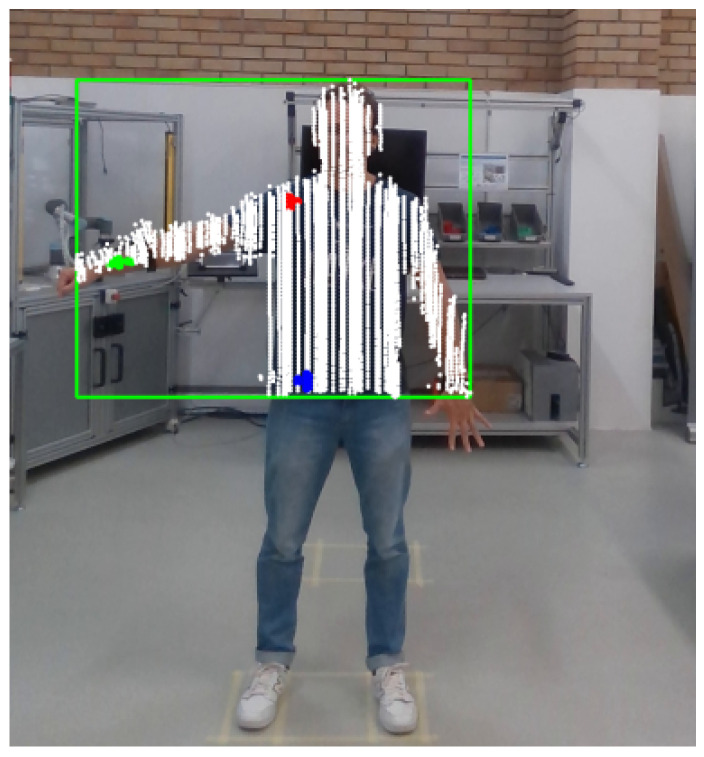
Example of the joints identified in the person. **Green points** related to the right wrist. **Red points** related to the right shoulder. **Blue points** related to the right hip.

**Figure 8 sensors-24-00530-f008:**
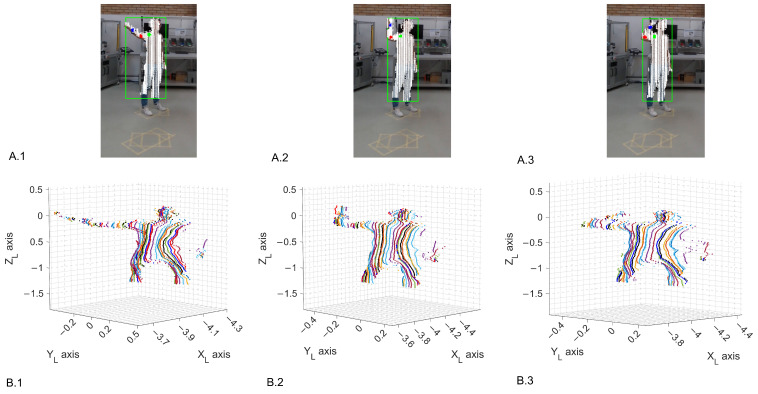
Example of joint identification, with the equivalent fitted point cloud for the elbow flexion ($R_{7}$) at a 4−m distance and an oblique view: (**.1**) at a 130° angle; (**.2**) at a 96° angle; (**.3**) at a 70° angle. (**A**) The points marked in green, red, and blue correspond to the shoulder, elbow, and wrist, respectively. (**B**) colors represents distinct data captured by the LiDAR at different iteration.

**Figure 9 sensors-24-00530-f009:**
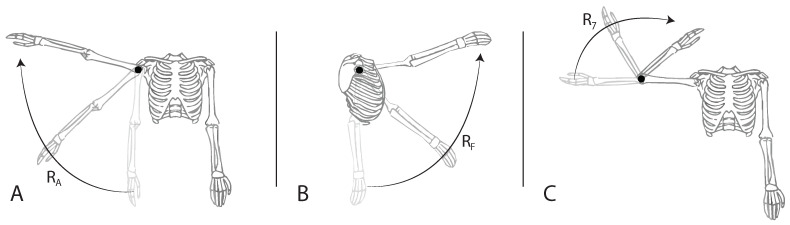
Demonstration of the three performed exercises in frontal view, with the targeted rotation joint. (**A**) Shoulder abduction ($R_{A}$), (**B**) shoulder flexion ($R_{F}$), (**C**) elbow flexion ($R_{7}$), and (•) targeted rotation joint.

**Figure 10 sensors-24-00530-f010:**
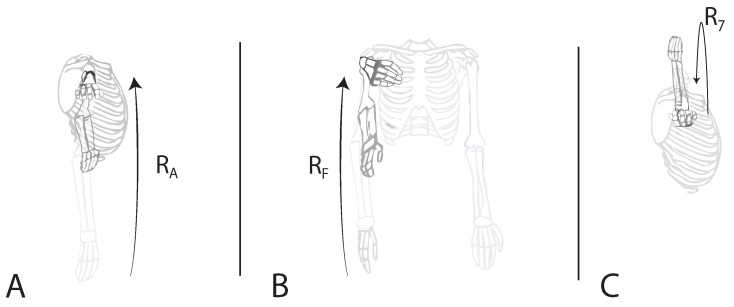
Demonstration of the three performed exercises in lateral view. (**A**) Shoulder abduction (*R_A_*), (**B**) shoulder flexion (*R_F_*), and (**C**) elbow flexion (*R*_7_).

**Figure 11 sensors-24-00530-f011:**
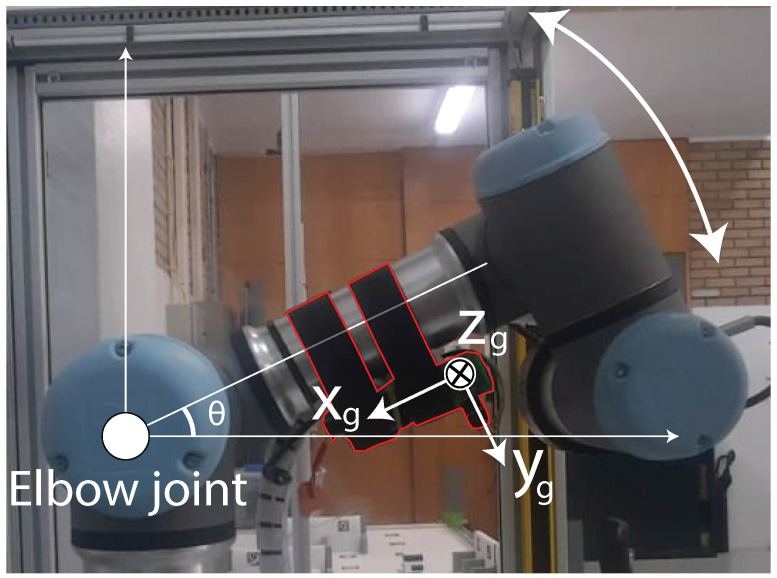
Wearable module attached to the UR3 with its reference coordinate frame and the coordinate frame selected for θ calculation.

**Figure 12 sensors-24-00530-f012:**
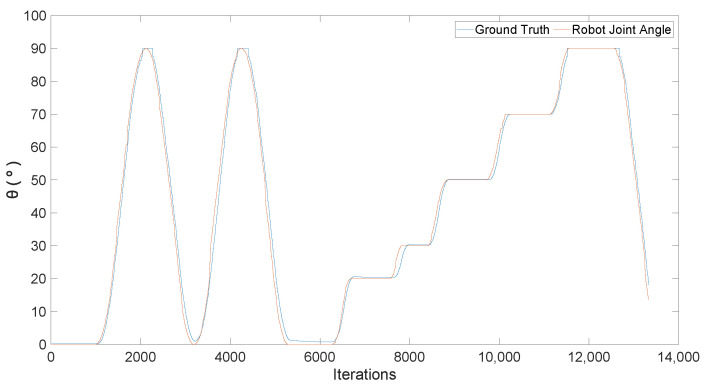
UR3 joint movements compared to the angle measured using the IMU.

**Figure 13 sensors-24-00530-f013:**
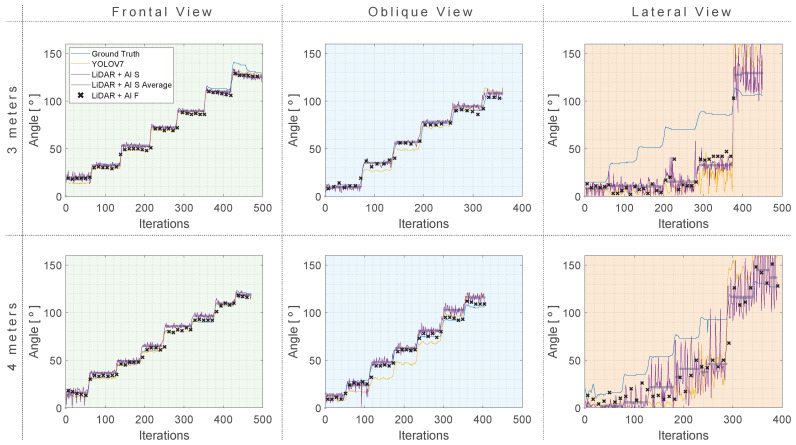
Shoulder abduction ($R_{A}$) results with three approaches compared to a ground truth reference at 4 and 3 m distances. (**Green**) Frontal View. (**Blue**) Oblique view. (**Orange**) Lateral view.

**Figure 14 sensors-24-00530-f014:**
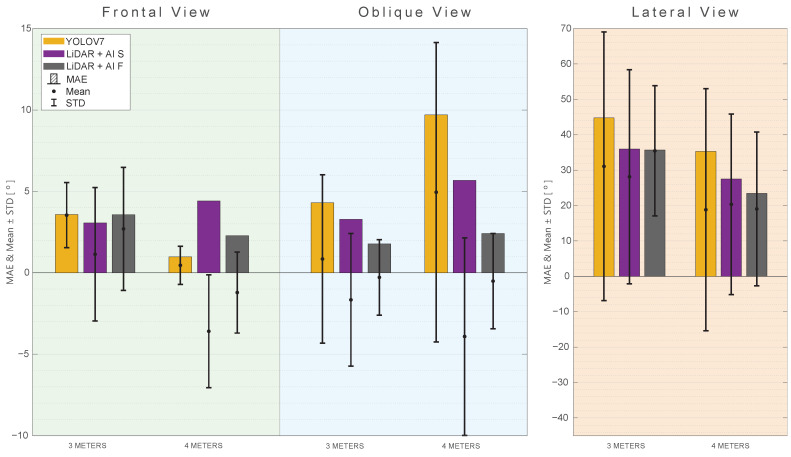
MAE, mean, and standard deviation for the $R_{A}$ rotation angle with different views and distances. (**Green**) Frontal View. (**Blue**) Oblique view. (**Orange**) Lateral view.

**Figure 15 sensors-24-00530-f015:**
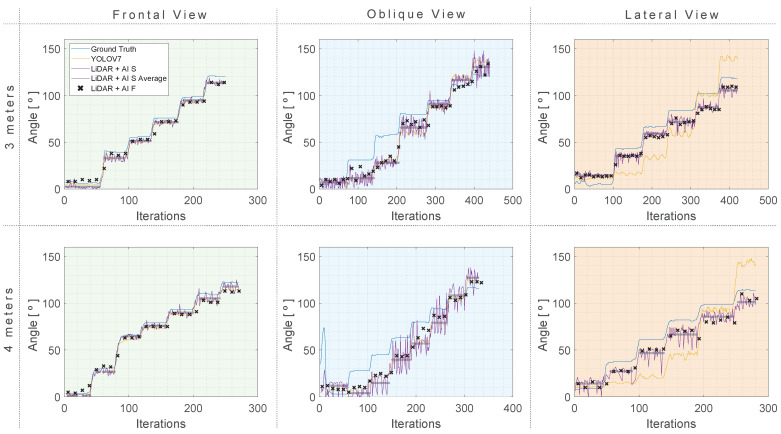
Shoulder flexion ($R_{F}$) results with three approaches compared to a ground truth reference at 4 and 3 m distances. (**Green**) Frontal view. (**Blue**) Oblique view. (**Orange**) Lateral view.

**Figure 16 sensors-24-00530-f016:**
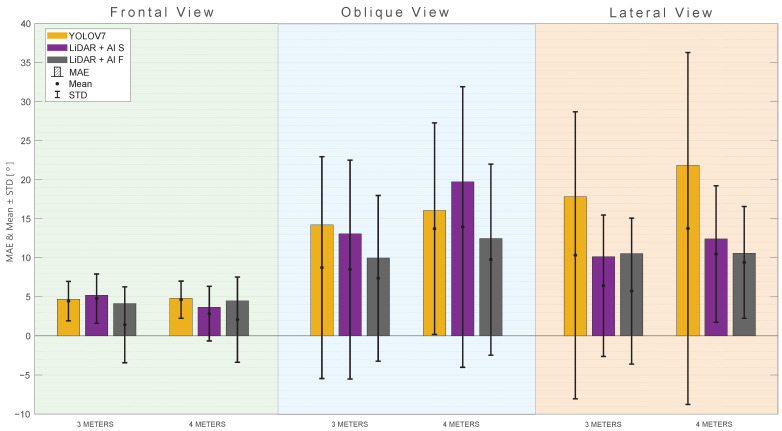
MAE, mean, and standard deviation for the $R_{F}$ rotation angle with different views and distances. (**Green**) Frontal view. (**Blue**) Oblique view. (**Orange**) Lateral view.

**Figure 17 sensors-24-00530-f017:**
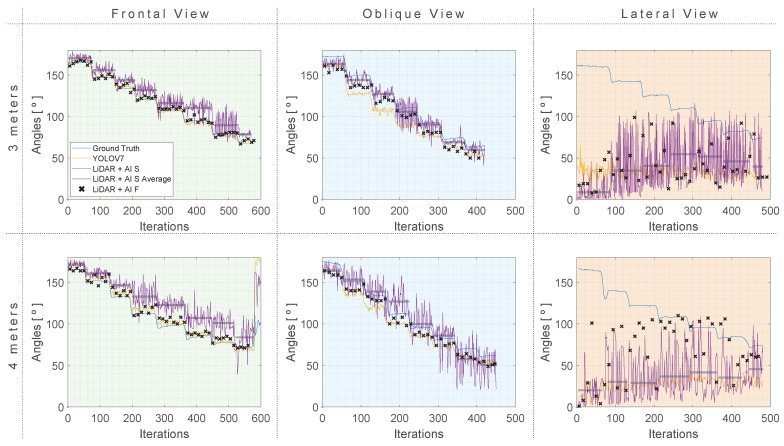
Elbow Flexion ($R_{7}$) results with three approaches compared to a ground truth reference at 4 and 3 m distances. (**Green**) Frontal view. (**Blue**) Oblique view. (**Orange**) Lateral view.

**Figure 18 sensors-24-00530-f018:**
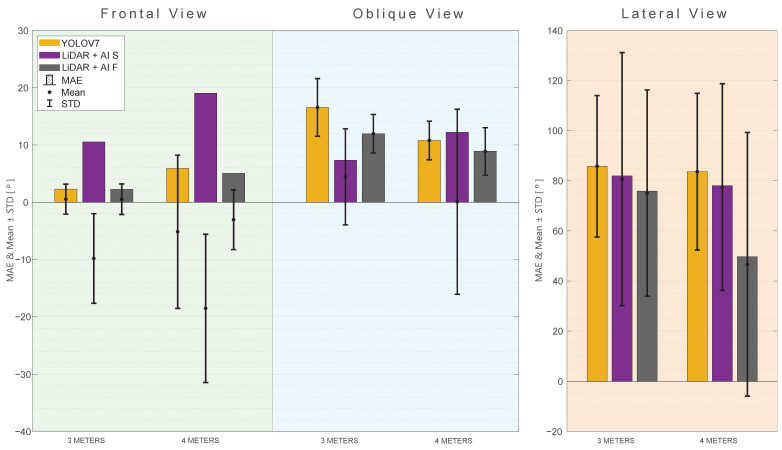
MAE, Mean, and Standard deviation for the $R_{7}$ rotation angle with different views and distances. (**Green**) Frontal view. (**Blue**) Oblique view. (**Orange**) Lateral view.

**Figure 19 sensors-24-00530-f019:**
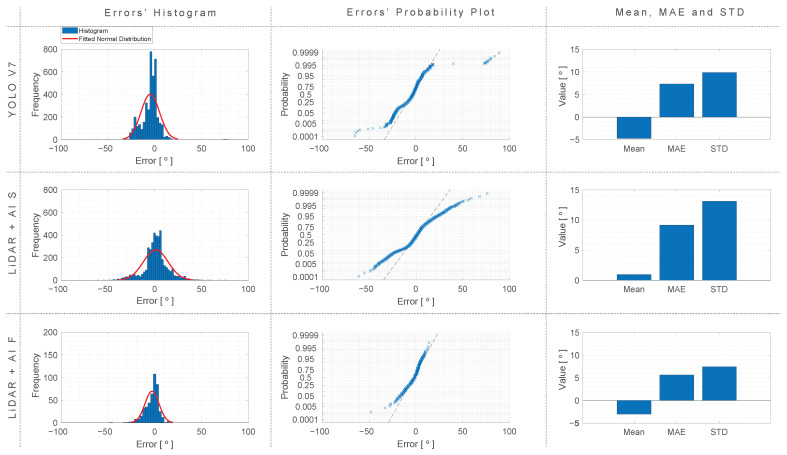
Histogram, probability plot, mean, MAE, and standard deviation of errors for the different approaches, computed with the data from all the exercises, views, and distances (excluding lateral view).

## Data Availability

Data are contained within the article.

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
