# Peer review of "Angle Assessment for Upper Limb Rehabilitation: A Novel Light Detection and Ranging (LiDAR)-Based Approach"

_sensors, 2024, doi:10.3390/s24020530_

Round 1

Reviewer 1 Report

Comments and Suggestions for Authors

The article is well-written and engaging, addressing an important and intriguing topic. I agree that there are many inaccuracies in the measurements of ranges, but these inaccuracies are known and have been well-documented and measured.

The authors propose a complex and detailed method for shoulder and elbow, clinically cumbersome and expensive, requiring multiple analyses and integrations of several systems. If we analyze the known difficulty and statistical error of measurements in goniometry or any motion analysis devices against the system presented here, the cost-benefit is unreasonable and does not promote clinicians in their daily clinical work. Furthermore, the proposed equipment is not mature enough for research.

Despite the well-written professional document, I don't recommend publishing it. This article may fit another journal.  

Author Response

Dear Reviewer,

Thank you for your insightful review and constructive feedback on our manuscript. We appreciate your recognition of the article's engaging nature and focus on a significant topic. We want to address your comments and further clarify our paper's contributions:

  1. Preliminary Nature and Analysis: As a preliminary study, our paper provides an extensive analysis of the proposed method, detailing its features, strengths, and challenges. This comprehensive examination aims to catalyze further research in the field, laying a foundation for future innovation.
  2. Validation of a Concept: Our work aims to validate an idea rather than present a market-ready product. We acknowledge the nascent stage of our proposed system and aim to foster development towards practical solutions, particularly in clinical applications.
  3. Exclusion of Cost-Effectiveness Analysis: Our study focuses on the theoretical and technical feasibility of the concept rather than its immediate commercial viability. As a preliminary investigation, we emphasize conceptual validation and theoretical development.
  4. Clarification of Objectives: Our paper presents an academic study and not a commercial product. Thus, if any part of our manuscript inadvertently suggests a finalized product, we are prepared to revise it for clarity.

In summary, our paper contributes to the field through its theoretical insights and methodological advancements, aligning with the journal's mission to promote academic discourse and innovation. We hope these clarifications address your concerns and demonstrate the value of our paper to the academic community. We respectfully request reconsideration of our manuscript for publication.

Reviewer 2 Report

Comments and Suggestions for Authors

The title and abstract do not make the object and objective of the study transparent.

In contextualizing the work, I believe that the authors need better arguments regarding the justification and relevance of the study. How viable is a complex (expensive?) device to make relatively simple measurements? The ROM measurement does not have this weight of importance for rehabilitation procedures.

I believe that the contextualization of the work and the discussion of the findings need to be better organized and presented.

Author Response

In contextualising the work, I believe that the authors need better arguments regarding the justification and relevance of the study. How viable is a complex (expensive?) device to make relatively simple measurements? The ROM measurement does not have this weight of importance for rehabilitation procedures.

Thank you for your pertinent question. We appreciate your thoughtful comment on the relevance of our study. As a preliminary exploration, the paper recognises the inherent limitations, including the complexity and potential expense associated with the proposed device.

It is essential to recognise that this work serves as a springboard, aiming to trigger further exploration in an area that has received less attention than other areas that have been extensively researched, such as the use of AI and IMU. As this is a preliminary study in an academic context rather than a commercial product development, factors such as price are currently outside the focus. However, we acknowledge and transparently communicate these considerations, which have been clearly stated in the conclusion section.

The value of this work lies in its potential to stimulate wider avenues of research by presenting a new approach to measuring range of motion (ROM) directly from the patient. Beyond clinical rehabilitation procedures, we envisage future validations that could extend the applicability of this technology, where there is an exciting prospect of using this technology in high-performance model training scenarios.

I believe that the contextualisation of the work and the discussion of the findings need to be better organised and presented.

Thank you for your comments. We take note of your suggestion concerning the contextualisation of the work and the organisation of the results. We value your ideas and are committed to improving these aspects based on your recommendations.

At this stage, without a clear definition or proposal of the specific areas or manner in which reorganisation is desired, we find it challenging to propose alternatives that would address the concerns raised by all relevant stakeholders.

We welcome further guidance and specific feedback to ensure the adjustments align with readers' expectations and preferences.

Reviewer 3 Report

Comments and Suggestions for Authors

This paper investigates the possibility of the use of a new approach to angle calculation: a hybrid approach leveraging both a camera and LiDAR technology, merging image data with point cloud information. This paper is logical, workload, clear thinking and innovative. However, the following problems still need attention and improvement:

2. Related Work

The introduction of existing joint angle prediction methods is too lengthy and requires textual refinement and summary. Or classify and introduce methods for predicting joint angles, such as 2.1 artificial intelligence and depth cameras, 2.2 IMU, and so on.

It is necessary to clarify the existing problems in previous research and the advantages of using LiDAR for joint angle estimation adopted in this article. Can it solve the unresolved problems in extracting joint angles using other methods.

3. Methodology

3.1. Data collection

For the signal collection section, does Ground Truth refer to the spatial position of the arm? Arm rotation angle? What is the distance parameter between the arm and the robotic arm? Further clarification is needed to clarify the specific meaning of the collected signal.

3.2. Data processing and angle calculation

It is recommended to provide a detailed description of each process in terms of angle processing, which will enhance the logical coherence. For example, 3.2.1 point cloud fusion of the point clouds, 3.2.2 person's pose collections for personnel pose collection, 3.2.3 identification of the person's identity, and so on.

3.3. Rehabilitation Exercises

This section only presents the joint changes of the arm in three view directions, and does not show the joints between arm rehabilitation training, which should belong to human biology. The research purpose of this article is also to identify and estimate joint angles, and the connection between rehabilitation training and joint changes is not very clear.

3.4. Ground Truth

Propose using IMU to compare and evaluate angle estimation results. But only the IMU and UR3 robotic arms were introduced, without explaining how to compare and evaluate angles? What angles of upper limb joints are compared and evaluated? What are the indicators for evaluation? What are the standards for indicators?

In addition, there are the following questions that need to be modified or answered:

(1) What is the basis for the horizontal movement of a robotic arm, and does it require trajectory tracking of human joints?

(2) The research method is based on the coordinate system of the robotic arm itself, and will the vibration caused by the trajectory movement of the robotic arm and the encoder cause systematic errors in the indicators of upper limb rehabilitation evaluation?

(3) The role and significance of using YOLO v7 for human identity recognition and upper body segmentation are unclear (it is possible to only match the joints of the upper body in the joint matching section without losing images such as hands).

(4) After identifying the clearance nodes, do you need to directly connect the joint points with curves or straight lines to demonstrate the accurate posture of the joints and bones.

(5) Compared with other image-based methods, this method has slight improvements in angle measurement. What impact do you think the cost of this method and the smoothness limitations in motion execution have on its application?

(6) How does the results of this study affect the accuracy of decision-making and angle measurement during patient rehabilitation?

Comments on the Quality of English Language

The language level should be improved through a careful reading of the paper as there are many minor English mistakes especially in the use of singular versus plural and verb tenses. As these are minor errors the paper can still be read and understood but they lower the overall quality of the work and should be eliminated.

Author Response

Review 3

This paper investigates the possibility of the use of a new approach to angle calculation: a hybrid approach leveraging both a camera and LiDAR technology, merging image data with point cloud information. This paper is logical, workload, clear thinking and innovative. However, the following problems still need attention and improvement:

Dear reviewer, Thank you very much for your comments and suggestions. Each of the points were carefully taken in consideration, and our answers to your questions are below.

  1. Related Work

The introduction of existing joint angle prediction methods is too lengthy and requires textual refinement and summary. Or classify and introduce methods for predicting joint angles, such as 2.1 artificial intelligence and depth cameras, 2.2 IMU, and so on.

Thank you very much for your recommendation. We fully agree with your suggestion and have accordingly included it. 

It is necessary to clarify the existing problems in previous research and the advantages of using LiDAR for joint angle estimation adopted in this article. Can it solve the unresolved problems in extracting joint angles using other methods

This is an interesting question, we have tried to keep the related work as objective as possible so as not to influence readers in general and to permit them to compare the method with others proposed in the literature.

The benefits lie mainly in highlighting the possibility of using such a device and encouraging further research in this area, as other methods have been extensively researched. This wireless and remote method offers superior accuracy to other remote methods for angles up to 45º, and without encumbering patients by wearing a device. 

We have worked on the whole article to make this observation more obvious to readers.

  1. Methodology

3.1. Data collection

For the signal collection section, does Ground Truth refer to the spatial position of the arm? Arm rotation angle? What is the distance parameter between the arm and the robotic arm? Further clarification is needed to clarify the specific meaning of the collected signal.

Thank you for your questions, they are essential for improving the content. 

We have performed modifications in Section 3.4 to clarify questions related to the Ground Truth system, which returns the angle of the arm or forearm, depending on the movement performed. 

The data was collected in three rotational movements of the shoulder and arm at a distance of 3 and 4 meters between the person and the camera in frontal, oblique and lateral views. The IMU sensor used was configured to return the direction of the gravity vector in relation to the sensor. This information is later processed to obtain the joint angulation. This method was validated according to the experiments using the robot specified in Section 3.4.

3.2. Data processing and angle calculation

It is recommended to provide a detailed description of each process in terms of angle processing, which will enhance the logical coherence. For example, 3.2.1 point cloud fusion of the point clouds, 3.2.2 person's pose collections for personnel pose collection, 3.2.3 identification of the person's identity, and so on.

We appreciate your inquiry concerning this specific aspect. Following your suggestion, we have implemented the separation of each part of the data processing into distinct subsections, to make a detailed description of the steps. We aimed to enhance coherence and improve the overall quality of the text.

3.3. Rehabilitation Exercises

This section only presents the joint changes of the arm in three view directions, and does not show the joints between arm rehabilitation training, which should belong to human biology. The research purpose of this article is also to identify and estimate joint angles, and the connection between rehabilitation training and joint changes is not very clear.

Thank you for your inquiry regarding this particular aspect.

The nexus between joint alterations and rehabilitation training lies in the fact that, during the execution of exercises integral to rehabilitation, joints play a pivotal role in carrying out movements. Consequently, besides other contributing factors, impairments or injuries affecting a patient's ability to perform natural movements often stem from limitations in joint mobility. Thus, changes in joints, as well as the angles formed by these changes, are inherently intertwined with the rehabilitation process.

To provide a concise overview of joints in rehabilitation training, we have included a brief explanation. More in-depth information can be explored in the referenced work (reference 17).

It is important to note that the current exercises specifically focus on the rotational aspect of joints during rehabilitation exercises. For example, the angle denoted as R7 corresponds to the elbow joint's rotation during the exercise known as elbow flexion. In the current contexts, both terms can be understood interchangeably.

3.4. Ground Truth

Propose using IMU to compare and evaluate angle estimation results. But only the IMU and UR3 robotic arms were introduced, without explaining how to compare and evaluate angles? What angles of upper limb joints are compared and evaluated? What are the indicators for evaluation? What are the standards for indicators?

Thank you for your inquiry regarding this particular aspect.

We have performed modifications in Section 3.4 aiming to clarify these aspects.

In addition, there are the following questions that need to be modified or answered:

(1) What is the basis for the horizontal movement of a robotic arm, and does it require trajectory tracking of human joints?

Thank you for your inquiry regarding this particular aspect. The primary objective behind the horizontal movement of the robot arm was to enhance data collection by obtaining more points from the LiDAR during the data collection process. The chosen LiDAR model, RS 16, lacked the necessary resolution to capture data when stationary adequately. Consequently, keeping the sensor static would result in the collection of only a limited amount of data.

To address this limitation and improve data resolution, we opted to move the LiDAR to different positions, and then merge the collections into one, as illustrated in Figure 4. By dynamically positioning the LiDAR, we sought to augment the resolution of the collected data. In essence, the deliberate movement of the LiDAR, facilitated by the robot arm whose precise position we could accurately determine, aimed to provide a more detailed understanding of the measurements.

It is important to note that the choice of the robot arm was pragmatic, as we had access to one and could precisely determine its position, thereby ensuring accurate LiDAR placement. However, alternative approaches could yield similar results. For instance, one could place the LiDAR in 11 predetermined positions with known coordinates or employ multiple LiDARs in varied orientations.

(2) The research method is based on the coordinate system of the robotic arm itself, and will the vibration caused by the trajectory movement of the robotic arm and the encoder cause systematic errors in the indicators of upper limb rehabilitation evaluation?

Thank you for your inquiry regarding this specific aspect. It is crucial to emphasize that while the movement of the robot was a key element in our research, it served merely as a tool to enhance LiDAR resolution. As discussed earlier, alternative methodologies could be employed to achieve similar outcomes.

Regarding the vibration of the robot arm, it is noteworthy that this aspect went unobserved in the system. This can be attributed to the deliberately slow velocity of the robot and its classification as an industrial robot, thereby ensuring a high level of precision. Considering these factors, coupled with the understanding that our research is a preliminary study and not a market-ready product, we deemed these considerations negligible. However, the authors acknowledge the importance of addressing this in future studies.

(3) The role and significance of using YOLO v7 for human identity recognition and upper body segmentation are unclear (it is possible to only match the joints of the upper body in the joint matching section without losing images such as hands).

Thank you for inquiring about this specific aspect. Similar to the utilization of the robot arm and the movement of the LiDAR, the adoption of YOLO v7 served as a means to validate our approach and can be substituted with other methodologies in subsequent efforts, with the ultimate goal of refining the study for market readiness.

Although crucial for automatic matching, the use of YOLO v7 was used aiming to be a solution to the challenges due to the distinct Fields of View (FoV) of the camera and the LiDAR. Direct adaptations were necessary, given the initial uncertainty about the correspondence between LiDAR points and individual joints. To address this, conversion of points into an "image" was required, followed by matching it to the person's body to enable automated identification. Then, the use of Yolo V7 to identify the person in the image was important.

It is important to note that this process has inherent limitations, and there exists a substantial opportunity to enhance the system's quality. While adequate for validating our approach, we recognize this as an aspect ripe for improvement in future endeavours, aligning to develop a final product suitable for market utilization.

(4) After identifying the clearance nodes, do you need to directly connect the joint points with curves or straight lines to demonstrate the accurate posture of the joints and bones.

Thank you for delving into this particular aspect. It presents an intriguing opportunity for integration into our approach, especially for physiotherapists. However, it is important to clarify that this feature is slated for inclusion in the market-ready product. In the current studies, focused on approach validation, such a feature was not deemed necessary. At this juncture, the responsibility for this aspect falls within the purview of the system operator and lies outside the scope of the proposed approach.

(5) Compared with other image-based methods, this method has slight improvements in angle measurement. What impact do you think the cost of this method and the smoothness limitations in motion execution have on its application?

Thank you for your inquiry into this specific aspect. We acknowledge the limitations inherent in the proposed approach, particularly concerning cost considerations. However, at this juncture in our research, cost is not a primary concern. The current focus is on the initial stages of our studies, with the main objective being the validation of our approach, showcasing a potential innovative method for angle calculation.

It is noteworthy that, to the best of our knowledge, no other studies have explored this particular avenue, positioning us as pioneers in this method. Furthermore, the authors recognize that future endeavours can build upon our proposed framework, addressing and refining the limitations, especially in terms of costs. Our immediate goal is to demonstrate the feasibility of the approach, thereby paving the way for subsequent studies that can enhance and optimize its practicality.

(6) How does the results of this study affect the accuracy of decision-making and angle measurement during patient rehabilitation?

Thank you for your interest in this specific aspect. A notable strength of this study, particularly in influencing decision-making during patient rehabilitation, lies in its integration with the SmartHealth Software (previously developed by the authors, references 11 and 12). This integration allows for the comprehensive tracking of patients' progress and maintains a detailed evolution history. Our system, with its high accuracy and automated data collection, eliminates the need for manual input by physiotherapists and can provide the history of the patient through time.

It is crucial to underscore that this preliminary study is exclusively focused on rehabilitation aspects. This serves as an opening for future research to refine and extend its applications to other domains. For instance, our system could prove beneficial in areas like high-performance training, despite the distinct nature of this field. The joint angle information generated by our approach could offer valuable insights into these diverse applications.

The language level should be improved through a careful reading of the paper as there are many minor English mistakes especially in the use of singular versus plural and verb tenses. As these are minor errors the paper can still be read and understood but they lower the overall quality of the work and should be eliminated.

We have reviewed the manuscript. Thank you for your suggestion.

Round 2

Reviewer 3 Report

Comments and Suggestions for Authors The authors have satisfactorily responded to my comments. Comments on the Quality of English Language

Moderate editing of English language required.